# Associations between Brain Alpha-Tocopherol Stereoisomer Profile and Hallmarks of Brain Aging in Centenarians

**DOI:** 10.3390/antiox13080997

**Published:** 2024-08-17

**Authors:** Jia Pei Chan, Jirayu Tanprasertsuk, Elizabeth J. Johnson, Priyankar Dey, Richard S. Bruno, Mary Ann Johnson, Leonard W. Poon, Adam Davey, John L. Woodard, Matthew J. Kuchan

**Affiliations:** 1Abbott Nutrition, Columbus, OH 43219, USA; matthew.kuchan@abbott.com; 2Friedman School of Nutrition Science and Policy, Tufts University, Boston, MA 02111, USA; elizabeth.johnson@tufts.edu; 3College of Education and Human Ecology, The Ohio State University, Columbus, OH 43210, USA; priyankar.dey@thapar.edu (P.D.); bruno.27@osu.edu (R.S.B.); 4Department of Biotechnology, Thapar Institute of Engineering & Technology, Patiala 147004, Punjab, India; 5Department of Nutrition and Health Sciences, University of Nebraska-Lincoln, Lincoln, NE 68588, USA; majohnson@unl.edu; 6Institute of Gerontology, University of Georgia-Athens, Athens, GA 30602, USA; lpoon8122@gmail.com; 7Department of Health Behavior and Nutrition Sciences, University of Delaware, Newark, DE 19716, USA; davey@udel.edu; 8Department of Psychology, College of Liberal Arts and Sciences, Wayne State University, Detroit, MI 48202, USA; john.woodard@wayne.edu

**Keywords:** vitamin E, tocopherol, brain, cognition, aging, neurofibrillary tangle, amyloid plaque, older adult, centenarian

## Abstract

Brain alpha-tocopherol (αT) concentration was previously reported to be inversely associated with neurofibrillary tangle (NFT) counts in specific brain structures from centenarians. However, the contribution of natural or synthetic αT stereoisomers to this relationship is unknown. In this study, αT stereoisomers were quantified in the temporal cortex (TC) of 47 centenarians in the Georgia Centenarian Study (age: 102.2 ± 2.5 years, BMI: 22.1 ± 3.9 kg/m^2^) and then correlated with amyloid plaques (diffuse and neuritic plaques; DPs, NPs) and NFTs in seven brain regions. The natural stereoisomer, *RRR*-αT, was the primary stereoisomer in all subjects, accounting for >50% of total αT in all but five subjects. %*RRR* was inversely correlated with DPs in the frontal cortex (FC) (ρ = −0.35, *p* = 0.032) and TC (ρ = −0.34, *p* = 0.038). %*RSS* (a synthetic αT stereoisomer) was positively correlated with DPs in the TC (ρ = 0.39, *p* = 0.017) and with NFTs in the FC (ρ = 0.37, *p* = 0.024), TC (ρ = 0.42, *p* = 0.009), and amygdala (ρ = 0.43, *p* = 0.008) after controlling for covariates. Neither *RRR-* nor *RSS*-αT were associated with premortem global cognition. Even with the narrow and normal range of BMIs, BMI was correlated with %*RRR*-αT (ρ = 0.34, *p* = 0.021) and %*RSS*-αT (ρ = −0.45, *p* = 0.002). These results providing the first characterization of TC αT stereoisomer profiles in centenarians suggest that DP and NFT counts, but not premortem global cognition, are influenced by the brain accumulation of specific αT stereoisomers. Further study is needed to confirm these findings and to determine the potential role of BMI in mediating this relationship.

## 1. Introduction

Supplementation studies in vitamin E-deficient humans have revealed that α-tocopherol (αT) is critical to nervous system function [1,2,3,4,5]. αT has the highest vitamin E activity of the eight vitamin E structural isomers (α-, β-, γ-, and δ-tocopherols and α-, β-, γ-, and δ-tocotrienols) and is the only one that can fulfill human vitamin E requirements [6]. The αT molecule contains three chiral carbons that give rise to eight possible stereoisomers (*RRR*-, *RRS*-, *RSR*-, *RSS*-, *SSS*-, *SSR*-, *SRS*-, and *SRR*-αT), but there is only one naturally occurring αT stereoisomer (*RRR*-αT; 2,5,7,8-tetramethyl-2 R-4′ R, 8′ R 12′ trimethyltridecyl)-6-chromanol). On the contrary, the αT commonly used in dietary supplements and fortified foods is synthetic αT (all-*rac*-αT), comprising eight stereoisomers in equimolar concentrations. Of these eight stereoisomers, only those in the *2R* configuration contribute to meeting human dietary vitamin E (αT) requirements.

Evidence shows that *RRR*-αT has 1.36–2 times the biological value of synthetic αT due to a stereoisomer selective hepatic transport protein, αT transfer protein (αTTP) [7,8,9,10]. While the human infant brain is known to accumulate *RRR*-αT preferentially, most infant decedents had measurable concentrations of synthetic αT stereoisomers in their brain, and some had as much as 48% of total αT [11]. While the distribution of αT stereoisomers in the adult brain has not been characterized, it is reasonable to expect that lifelong dietary exposure to natural and synthetic αT could result in the differential accumulation of brain αT stereoisomers.

Aging presents a tremendous global challenge, with demographic shifts toward an aging population imposing diverse health burdens. Central to these concerns is the imperative to maintain healthy brain aging, as the aging brain undergoes significant changes, often marked by neurofibrillary tangles (NFTs) and amyloid plaques, including diffuse plaques (DPs) and neuritic plaques (NPs) [12,13,14]. While traditionally associated with Alzheimer’s disease (AD), recent evidence indicates their presence in cognitively healthy older individuals [12,15,16,17]. Research suggests that reducing or clearing these lesions does not consistently lead to clinical improvement [13]. Clinico-neuropathological studies in the oldest old populations across different geographical regions reinforce this complexity by showing that NFTs and amyloid plaques alone cannot fully explain cognitive impairment [18,19,20,21]. For example, similar distributions of AD-type neuropathology among centenarians with varying cognitive statuses were observed in the Georgia Centenarian Study (GCS) [22]. Accumulating evidence challenges the conventional belief that NFTs and amyloid plaques are exclusive to AD and contribute directly to cognitive dysfunction. It also suggests the need to reevaluate the role of NFTs and amyloid plaques as markers in normal aging processes.

Nutritional intervention is an effective way to maintain healthy brain aging [23,24,25,26,27,28]. In particular, diets high in nuts and seeds, plant oils (such as olive oil), and green leafy vegetables are associated with a reduced risk of cognitive decline [29,30,31,32,33,34]. These foods are excellent sources of different nutrients, including naturally occurring *RRR*-αT. Other prominent sources of αT include fortified foods and dietary supplements. Research indicates that among US adults regularly taking nutritional supplements, αT intake from supplements is roughly 11 and 16 times greater than from dietary sources in men and women, respectively [35]. Specific to αT, its increased intake and higher serum αT concentrations have also been associated with a lower risk of developing AD later in life [36,37,38,39,40,41,42]. However, as most studies do not distinguish among the bioaccumulation of specific αT stereoisomers or between natural and synthetic sources of αT, a knowledge gap exists in understanding their differential effects on health [8]. Since αT stereoisomers in the adult brain have not been characterized, the impact of different stereoisomers on markers of the aging brain and cognitive health remains unexplored in humans.

The GCS provided an opportunity to address these research questions preliminarily since subjects donated their brain tissues upon death. These tissues were previously utilized to assess NFTs and amyloid plaques and for αT analysis. Without differentiating among the stereoisomers, an earlier study reported that brain αT concentrations were positively associated with specific premortem cognitive tests, but not global cognition, in the GCS cohort [43]. In addition, a recent study reported that brain αT concentration was associated with reduced amyloid plaques and NFTs [44]. In this present study, we aimed to expand the scope of the investigation to establish the relationship of specific αT stereoisomers in the brains of older adults with global cognitive status, NFTs, and amyloid plaques. We hypothesized that differential accumulation of specific αT stereoisomers in the human brain, particularly the naturally occurring *RRR*-αT, may be more effective in maintaining cognitive function and associated with fewer NFTs and amyloid plaques. This study was therefore designed to provide a basis for further exploration of the potential for αT to achieve healthy aging and better maintain cognitive function.

## 2. Materials and Methods

### 2.1. GCS Decedents, Demographic Data, and Brain Collection

The University of Georgia Institutional Review Board on Human Subjects approved the GCS study. Separate approval to use de-identified samples and data for the analyses was obtained from the Tufts University/Tufts Medical Center Institutional Review Board (#8900). The GCS study was a population-based study conducted in a 44-county area in northeast Georgia, USA. The design, recruitment, and brain sample collection procedure have been described elsewhere [45,46].

In addition to collecting demographic data and blood samples, the Global Deterioration Scale (GDS, an assessment of global cognition) [47] was assessed at enrollment and approximately every six months after that at the subject’s residence until mortality, as previously described [46]. Only the blood samples and GDS from the final assessment, which was within one year of mortality for all subjects, were used in the analysis in this report. Serum samples were stored at −80 °C until analysis. Premortem habitual intake of αT could not be calculated since the intake assessment performed in the GCS did not provide sufficiently detailed information [48].

### 2.2. Brain and Serum Tocopherol Analysis

Brain αT and γ-tocopherol (γT) extraction and αT stereoisomer analyses were performed in a blinded manner at the Ohio State University (Columbus, OH, USA) as previously described [49,50]. Briefly, brain tissue (temporal cortex, TC) was saponified (30 min, 70 °C) in alcoholic potassium hydroxide and extracted with hexane. A portion of the hexane extract was dried under nitrogen gas, reconstituted in methanol/ethanol (1:1, *v*/*v*), and injected on an HPLC with electrochemical detection (HPLC-ECD) to determine total αT and γT as described [50]. To determine the percent distribution of αT stereoisomers, a separate portion of the hexane extract was dried under nitrogen gas, reconstituted, and subjected to methylation under basic conditions. The methylated sample was injected into an HPLC-fluorescence system with a chiral separation column. Under these conditions, a single peak encompassing total *2S*-αT stereoisomers and individual peaks for each *2R* stereoisomer of αT (*RRR*-, *RRS*-, *RSR*-, and *RSS*-αT) could be determined. The molar concentration of each αT stereoisomer was determined based on their percent distribution relative to the total αT established by HPLC-ECD.

The methods for measuring serum total αT and γT concentrations have been previously described, and the results were reported in a previous study [51]. αT stereoisomers were not measured in the serum.

### 2.3. Diffuse Plaque, Neuritic Plaque, and Neurofibrillary Tangle Assessment

After the autopsy, markers for aging brain assessment in the GCS included DPs, NPs, and NFTs. The assessment was previously described by Neltner et al. [21]. Briefly, DP, NP, and NFT counts were averaged from five microscopic fields that were most severely affected in each section from the following brain regions: the frontal cortex (FC, Brodmann Area 9), TC (Brodmann Areas 21–22), parietal cortex (Brodmann Areas 39–40), amygdala, entorhinal cortex, hippocampus section CA1, and subiculum.

### 2.4. Statistical Analysis

All analyses were performed in 47 subjects, except for any analyses that considered BMI (data not available in one double amputee) or markers of brain aging (data not available in three subjects, none of whom were the double amputee). Continuous data are expressed as mean ± standard deviation when the distribution is normal or as median [interquartile range] when the distribution is not normal. Categorical data are expressed as count (%). Spearman’s correlation and partial correlation analyses were used to examine the correlations between brain tocopherol concentrations and markers of brain aging, as well as between brain tocopherol concentrations and continuous demographic data (such as BMI). The Wilcoxon rank-sum test was used to examine the correlations between brain tocopherol concentrations and categorical demographic data. Statistical significance was set at α = 0.05 (*p* < 0.05). Given the nature of an exploratory study, correlations with *p* < 0.10 were also mentioned but not considered statistically significant. No adjustment of *p* values for type I errors due to multiple comparisons was performed. All tests were performed using R software (version 4.1.0 accessed on May 18, 2021, https://www.R-project.org).

## 3. Results

### 3.1. Cohort Description

Brain tissues from the TC of 47 subjects in the GCS were available for the analysis of tocopherol concentrations. Their characteristics are described in Table 1. Overall, subjects were 102.2 ± 2.5 years old, and the majority were female (89%), Caucasian (89%), and had a normal body mass index (BMI, 22.1 ± 3.9 kg/m^2^). The subjects were almost equally distributed for the absence (49%) or presence (51%) of dementia based on the GDS.

### 3.2. Temporal Cortex αT and γT and αT Stereoisomer Distribution

The TC distributions of αT and γT in 47 subjects are displayed as pmol/mg tissue (Appendix A) and as relative concentrations (% of total tocopherol, Appendix A). The concentrations were not normally distributed among the subjects. TC αT was 16.84 pmol/mg [9.87–22.70], and γT concentration was 0.16 pmol/mg [0.08–0.32]. αT was the predominant form of tocopherol in the TC of all subjects (99.06 [97.72–99.61]%). γT was not detected in the TC of six subjects.

The TC αT stereoisomer concentrations are shown as pmol/mg tissue (Figure 1A) and relative abundance (% of total αT, Figure 1B). TC concentrations (pmol/mg) of αT stereoisomers were as follows: *RRR*-αT, 9.75 [5.71–14.81]; *RRS*-αT, 2.21 [1.17–3.14]; *RSS*-αT, 1.80 [1.02–2.80]; *RSR*-αT, 1.56 [0.93–2.47]; and *2S*-αT (includes *SSS*-, *SSR*-, *SRS*-, and *SRR*-αT), 0.25 [0.13–0.43]. *RRR-*αT was the predominant stereoisomer in the TC of all subjects, accounting for >50% of total αT concentration in all but five subjects. The *RRR*-αT stereoisomers accounted for 37.78–48.52% of total αT in these five subjects.

TC γT concentration (pmol/mg) was not significantly correlated with the concentration of any of the αT stereoisomers in the TC (Appendix A). In contrast, each αT stereoisomer (pmol/mg) was positively correlated with all other αT stereoisomers (*p* < 0.001 for all). Spearman’s ρ and *p* values are listed in Appendix A.

### 3.3. Temporal Cortex Tocopherol and Global Deterioration Scale (GDS)

Neither the TC concentrations of αT nor γT nor the percent distribution of αT stereoisomers significantly differed between subjects with and without dementia (GDS1–3 and GDS4–7, Appendix A). Likewise, there were no significant correlations between concentrations or percentages of αT stereoisomers in the TC with the GDS examined as a continuous scale.

### 3.4. Temporal Cortex Tocopherol and Diffuse Plaques, Neuritic Plaques, and Neurofibrillary Tangles

We examined correlations between TC αT stereoisomers and DP, NP, and NFT counts in each brain region. After adjusting for sex, race, education, ApoE genotype, diabetes, and hypertension, TC %*RRR*-αT was inversely correlated with DP counts in the FC (ρ = −0.35, *p* = 0.032) and TC (ρ = −0.34, *p* = 0.038, Figure 2). Correlations between TC %*RRR*-αT and NFT counts did not reach significance in the FC (ρ = −0.30, *p* = 0.073), TC (ρ = −0.30, *p* = 0.070), amygdala (ρ = −0.31, *p* = 0.066), or hippocampus (ρ = −0.29, *p* = 0.087).

In contrast, TC %*RSS*-αT was positively correlated with DP counts in the TC (ρ = 0.39, *p* = 0.017) but not in the parietal cortex (ρ = 0.30, *p* = 0.075) or entorhinal cortex (ρ = 0.29, *p* = 0.079). TC %*RSS*-αT correlations with NP counts did not reach significance in the amygdala (ρ = 0.29, *p* = 0.085) or entorhinal cortex (ρ = 0.32, *p* = 0.057). TC %*RSS*-αT was positively associated with NFT counts in the FC (ρ = 0.37, *p* = 0.024), TC (ρ = 0.42, *p* = 0.009), and amygdala (ρ = 0.43, *p* = 0.008), but did not reach significance in the parietal cortex (ρ = 0.31, *p* = 0.065) or subiculum (ρ = 0.33, *p* = 0.056). Appendix A presents the other stereoisomers, which were observed to be minimally correlated with markers of AD pathologies in the various brain regions.

### 3.5. Temporal Cortex Tocopherols and BMI

BMI was negatively correlated with TC total αT concentration (ρ = −0.32, *p* = 0.030) but not with brain γT concentration (Appendix A). This correlation remained statistically significant after adjusting for total αT serum concentrations (ρ = −0.32, *p* = 0.032). After adjusting for sex, race, and serum concentrations, the correlation became borderline significant (ρ = −0.29, *p* = 0.059).

We next considered whether BMI was related to specific TC αT stereoisomers. A negative correlation was observed between BMI and the TC concentrations of *RRS*-αT (ρ = −0.36, *p* = 0.013), *RSS*-αT (ρ = −0.49, *p* < 0.001), and *RSR*-αT (ρ = −0.35, *p* = 0.017). In contrast, no significant relationship was observed with *RRR*-αT or *2S*-αT concentrations (Appendix A). The same pattern was observed after adjusting for sex and race (*p* < 0.05 for all).

A similar relationship was observed between BMI and the TC percent distribution of each αT stereoisomer (Appendix A). As shown in Figure 3, we observed that BMI was positively correlated with %*RRR*-αT (ρ = 0.34, *p* = 0.021) but was negatively associated with %*RSS*-αT (ρ = −0.45, *p* = 0.002). After adjusting for sex and race, these correlations remained statistically significant. The relative concentrations of other αT stereoisomers were not correlated with BMI either before or after adjusting for race and sex (*p* > 0.05).

Given that BMI was negatively associated with NFT and amyloid plaque counts in multiple brain regions (Appendix A), a partial correlation analysis adjusted for BMI in addition to sex, race, education, *ApoE* genotype, diabetes, and hypertension was also performed (Figure 2). Consequently, neither %*RRR*-αT nor %*RSS*-αT continued to be significantly correlated with AD pathology in any brain region after the inclusion of BMI in the partial correlation analysis (*p* > 0.10 for all).

## 4. Discussion

This study is the first to examine αT stereoisomer concentrations in human adult brain tissue and test their relation to premortem global cognitive status (GDS) and NFT and amyloid plaque counts. We report that *RRR*-αT, the naturally occurring stereoisomer, was the primary stereoisomer in the TC of all 47 GCS subjects and accounted for >50% of total αT in all but five subjects. While we found no relationship between TC αT stereoisomers and premortem GDS, *RRR*-αT and *RSS*-αT were differentially related to the measures of brain aging, with %*RRR*-αT inversely related to DPs in both the TC (ρ = −0.34, *p* = 0.038) and FC (ρ = −0.35, *p* = 0.032), but %*RSS* was positively correlated with both DPs in the TC (ρ = 0.39, *p* = 0.017) and with NFTs in the FC (ρ = 0.37, *p* = 0.024), TC (ρ = 0.42, *p* = 0.009), and amygdala (ρ = 0.43, *p* = 0.008). Stereoisomers share identical molecular formulas but differ only in three-dimensional shape. Therefore, our observation of opposing correlations with markers of brain aging in the human brain is surprising. However, the inclusion of BMI in the partial correlation analysis eliminated these associations.

We have previously reported that higher total αT was correlated with lower NFT counts in these GCS centenarians [44], but αT stereoisomer profiles were not characterized in that study. This current analysis clarified that opposing stereoisomer correlations exist beneath the inverse relation previously observed with total αT. The Memory and Aging Project also examined αT status and markers of brain aging in older adults. It revealed that αT was positively associated with amyloid load when γT levels were low, but was negatively associated with amyloid levels when γT levels were high [52]. The authors also suggested that αT created an anti-inflammatory environment in the brain and reduced total microglia density independent of amyloid load and NFT severity [53]. While associations with markers of brain aging were found, both previous and current studies showed no relationship between total αT or αT stereoisomer profiles and cognitive performance based on Global Deterioration Scores (GDSs) [47]. This contrasts the findings of others who observed a significant relationship between dietary αT intake and cognition [36,37,38,39]. However, none of these prior studies considered the stereoisomer form of αT in the diet or in brain tissue. Therefore, the specific impact of each αT stereoisomer on cognitive health remains unknown. Additionally, since the reference ranges for brain αT or αT stereoisomer concentration among centenarians have not been established, our findings may reflect a ceiling effect such that the older adults may have already plateaued in their αT needs and cognitive health despite the presence of markers of brain aging.

Recent research has examined the biochemical and clinical disparities among αT stereoisomers, revealing the body’s discrimination between *RRR*-αT and synthetic or all-*rac*-αT [11,54,55,56,57]. Administration of a mixed dose of *RRR*-αT and *SRR*-αT orally resulted in a preferential enrichment of *RRR*-αT in very-low-density-lipoprotein particles compared to *SRR*-αT [58]. Several mechanisms may elucidate these distinctions. Firstly, αT stereoisomers likely exhibit varying binding affinities to proteins, notably αTTP, pivotal in αT trafficking in the brain. Disruptions in this pathway can lead to ataxia [59,60]. Another protein, αT-associated protein (TAP), sharing the same binding site sequence with αTTP, serves as a transcriptional activator [61] and influences gene expression in the brain [49,62]. Additionally, αT stereoisomers might possess distinct antioxidant effects, impacting oxidative stress levels [63]. Similar steric structure–antioxidant activity relationships have been noted in tea catechins, where specific stereoisomers excel in scavenging large free radicals. In contrast, others are more efficient against smaller free radicals due to increased steric hindrance [64]. These antioxidant disparities could contribute to their roles in preventing lipid peroxidation, which is crucial in the brain due to its high polyunsaturated lipid content, which is susceptible to peroxidation with aging [65,66,67,68]. Thus, understanding the differential free radical-scavenging abilities of αT stereoisomers is crucial, given αT’s prominence as a lipid antioxidant in the brain. The lack of specificity in αT stereoisomers used in human clinical trials might explain the failure of interventions to improve cognitive measures in older adults at risk of cognitive impairment [30,69,70]. However, there are no available data that investigate and support these mechanisms of action in extreme aging. Therefore, future research should differentiate between *RRR*-αT and all-*rac*-αT and explore their associations with cognitive health in aging [8].

In the present study, we performed a correlation analysis to consider the influence of several covariates (e.g., sex, race, education, ApoE genotype, diabetes, and hypertension) that may affect the relationship between α-T stereoisomer profile and cognitive health. BMI was additionally included as a covariate in a secondary statistical model, given that BMI was positively associated with cognitive function in a cohort of healthy male veterans (aged 85 years and older, BMI 25.7 ± 3.2 kg/m^2^) [71], and that BMI was associated with both TC %*RRR*-αT and %*RSS*-αT in the GCS centenarians. Interestingly, the inclusion of BMI in the model resulted in the loss of statistical significance for the relationship between TC αT stereoisomers and brain aging biomarkers. Although this finding implies that BMI is confounding for this relationship, its appropriateness for inclusion in the statistical model is potentially lacking clinical relevance for our study population of centenarians. Indeed, BMI typically decreases with age in the oldest old, and it has also been utilized as a crude indicator of overall health and nutrition status [72,73]. In the ASPREE study [74], a strong U-shaped relationship was observed between BMI and all-cause mortality; for example, those classified as overweight (BMI 25–30 kg/m^2^) had a lower mortality risk compared to those with lower or higher BMI. Further, our cohort of centenarians had BMI values indicative of a range conventionally considered as healthy (22.1 ± 3.9 kg/m^2^) and showed limited inter-individual variability (CV = 17.6%) that would be expected to limit its usefulness as a covariate. Consistent with BMI indicative of being overweight (25–30 kg/m^2^, HR = 0.90, 95% CI = 0.85–0.95) or of obesity I (30–35 kg/m^2^, HR = 0.98, 95 CI = 0.92–1.06) being not or limitedly associated with mortality risk in older adults (70–79 y) [75], we speculate that its inclusion as a covariate in the present study is lacking relevance for our centenarian population, most of whom are normal weight and generally healthy. In addition, the variance in BMI in this study population was markedly narrower than that in the αT stereoisomers. It is possible that higher BMI and %*RRR*-αT both reflect a healthier lifestyle. For example, a diet rich in *RRR*-αT, found in natural foods like nuts, dark-green leafy vegetables, and egg yolk [6], would likely influence dietary and serum αT stereoisomer profiles in humans. In addition, lower BMI could predispose individuals to frailty, leading to increased consumption of health supplements containing synthetic αT, thus explaining the observed negative correlation between BMI and synthetic %*RSS*-αT. Additional study is warranted to evaluate the direct impact of overweightness and body composition on the accumulation of brain αT stereoisomers and their relationship with markers of brain aging to establish causal evidence to support healthy aging. 

The limitations of our findings should be mentioned. While the NFT, DP, and NP assessments were performed in different brain regions, only the TC was available to measure the αT stereoisomer profile. However, Kuchan et al. revealed that the concentrations of total αT and the proportions of *RRR*-αT and synthetic αT were similar among the infant brain regions examined, including the FC, hippocampus, and visual cortex, with *RRR*-αT being the most predominant form [11]. *RRR*-αT and synthetic αT proportions in six areas of infant rhesus macaque brains (occipital, temporal, motor, and prefrontal cortices together with the striatum and cerebellum) were also very similar within each treatment group [50]. As far as we know, there are no other studies that investigated αT and αT stereoisomer profiles in different brain regions in adult humans. Based on these findings, we believe extending the stereoisomer profile determined from the TC to other brain regions is reasonable. Moreover, high correlations among FC, TC, and circulating total αT and γT concentrations were previously reported in the GCS subjects [51]. That said, a study that investigates αT stereoisomer profiles in different brain regions in adult humans would be valuable to further confirm this assumption. Another limitation is extrapolation to other aged populations due to the small sample size and the unusual health status of the subjects. Lastly, while we did not perform any adjustment for type I error inflation, the correlations with BMI, NFTs, and DPs were consistent and specific to *RRR-*αT and *RSS-*αT.

## 5. Conclusions

This exploratory study provides a foundation for future research examining the impact of αT stereoisomers on cognitive function in aging. *RRR*-αT was the predominant αT stereoisomer in the TC of the oldest old. No significant difference in the αT stereoisomer profiles between subjects with and without dementia was observed. However, the proportion of *RRR*-αT was negatively and *RSS*-αT (a synthetic stereoisomer) was positively correlated with markers of brain aging (NFTs and DPs) in multiple brain regions. Although BMI confounded these associations, its clinical importance in a relatively healthy centenarian population may be confounding in itself. Further study through prospective trials are therefore needed to advance an understanding of our novel findings of αT stereoisomer accumulation in the centenarian brain relative to markers of brain aging with potential mediation by BMI status.

## Figures and Tables

**Figure 1 antioxidants-13-00997-f001:**
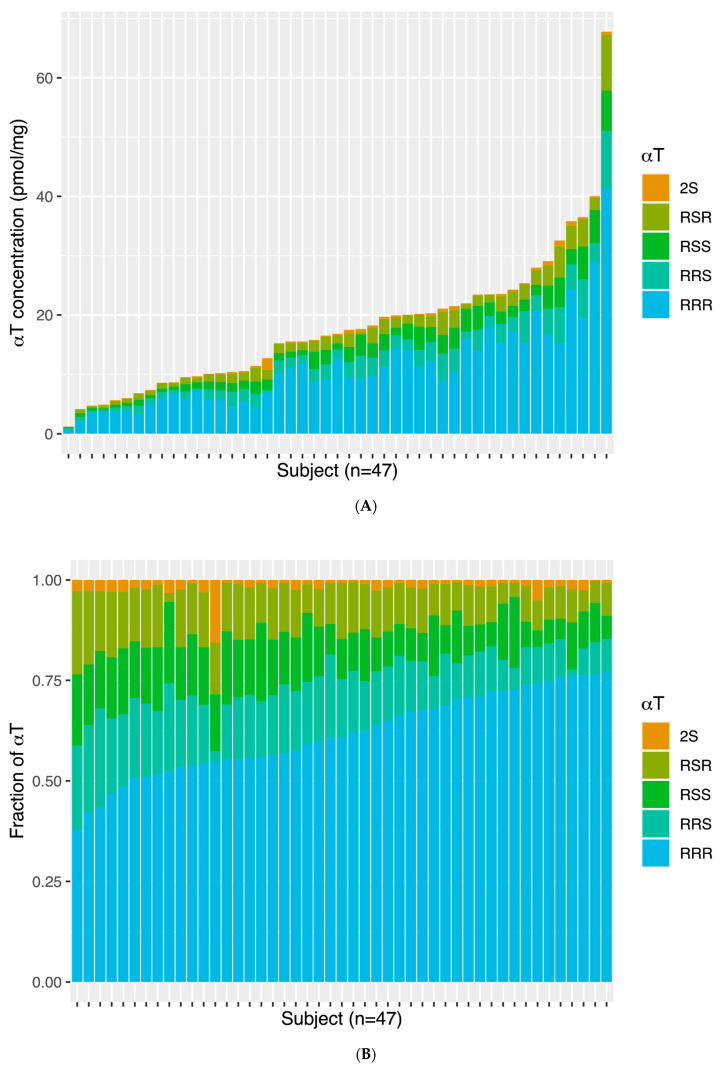
(**A**) α-tocopherol (αT) stereoisomer concentrations (pmol/mg tissue) in the brains of 47 subjects. Each bar represents a subject and subjects are ordered from the lowest to the highest total α-TP concentration. (**B**) αT stereoisomer relative concentrations in the brains of 47 subjects. Each bar represents a subject and subjects are ordered from the lowest to the highest %RRR, and 2S includes *SSS*, *SSR*, *SRS*, and *SRR*.

**Figure 2 antioxidants-13-00997-f002:**
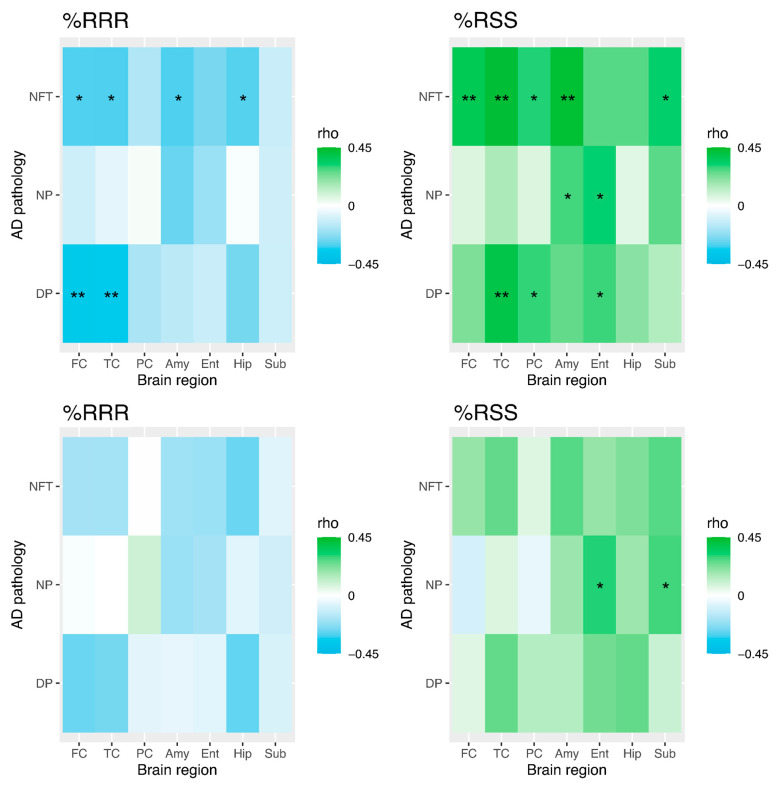
Correlation between %*RRR* or %*RSS* and diffuse plaque (DP), neuritic plaque (NP), or neurofibrillary tangle (NFT) counts in different brain regions (*n* = 43, excluding one double amputee and three without pathology assessment data). Partial correlation adjusting for sex, race, education, ApoE genotype, diabetes, and hypertension (**upper row**). Partial correlation adjusting for body mass index in addition to the variables in the first model (**lower row**). * *p* < 0.10 and ** *p* < 0.05.

**Figure 3 antioxidants-13-00997-f003:**
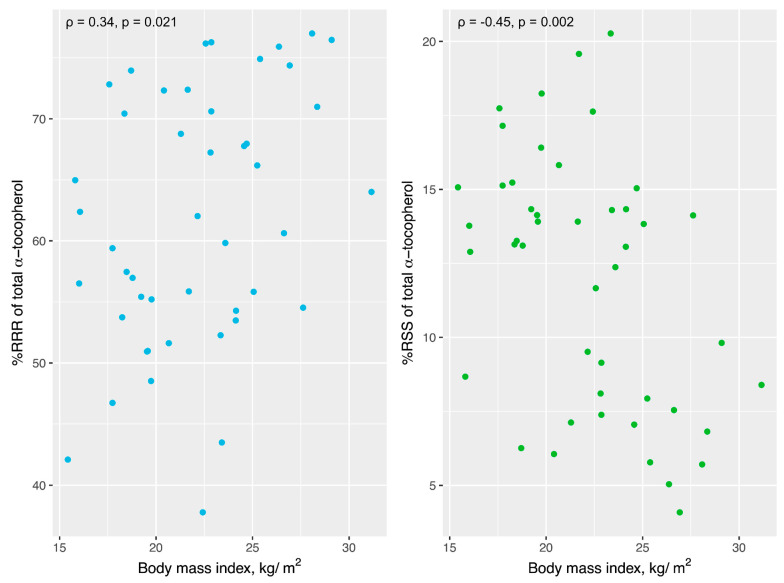
The relationship between brain tocopherol concentrations (% of total α-tocopherol) and body mass index (*n* = 46, excluding one double amputee). *RRR* (**left**) and *RSS* (**right**).

**Table 1 antioxidants-13-00997-t001:** Subject characteristics (*n* = 47).

Characteristic	Mean ± SD or Count (%)
Age, in years	102.2 ± 2.5
Sex	
Male	5 (11%)
Female	42 (89%)
Race	
Caucasian	42 (89%)
Black	5 (11%)
Body mass index, in kg/m^2^	22.1 ± 3.9
Education	
Lower than high school	23 (51%)
High school	12 (27%)
Higher than high school	10 (22%)
No data	2
Residence	
Community dwelling	33 (70%)
Institutionalized	14 (30%)
Diabetes	3 (6%)
Hypertension	25 (53%)
Alcohol status	
Never	21 (60%)
Past	6 (17%)
Present	8 (23%)
No data	12
Smoking status	
Never	30 (86%)
Past	4 (11%)
Present	1 (3%)
No data	12
Apo E genotype	
ε3/ε3	32 (68%)
ε2/ε3	7 (15%)
ε3/ε4	7 (15%)
ε2/ε4	1 (2%)
Global Deterioration Scale	
1–3 (no dementia)	23 (49%)
4–7 (dementia)	24 (51%)

Body mass index could not be calculated in one double amputee.

## Data Availability

The original contributions presented in the study are included in the article/Appendix A, further inquiries can be directed to the corresponding authors.

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
