# Peer review of "Associations between Brain Alpha-Tocopherol Stereoisomer Profile and Hallmarks of Brain Aging in Centenarians"

_antioxidants, 2024, doi:10.3390/antiox13080997_

Round 1

Reviewer 1 Report

Rather than studying total αT, this manuscript reports on investigations of the specific alpha tocopherol (αT) stereoisomers (RRR-αT and RSS-αT) which provides deeper information about thie antioxidant. This focus on stereoisomers has novelty, as previous studies have often treated αT as a homogenous compound without differentiating between its stereoisomers.

The study appears interesting, and it has potential novelty regarding alpha-tocopherol (αT) content of the aging brain. It explores the relationships between these stereoisomers and markers of brain aging (in particular neurofibrillary tangles and amyloid plaques), which has not been extensively studied before. The study uses brain samples from centenarians (subjects aged ≥98 years), which is a unique and valuable population for studying extreme aging. It also includes detailed analysis of body mass index (BMI). Although the design could not determine a relative importance of the stereoisomers concerning localization with neurofibrillary tangle and amyloid plaque counts, the study in my view provides novelty.

Overall, the manuscript provides a compelling and novel contribution to the field of the aging brain and vitamin E research, particularly through its focus on αT stereoisomers and their distinct roles in the aging brain.

Potential limitations:
i) The manuscript by design has a cross-sectional nature, which naturally limits the ability to determine causality. A longitudinal approach to examine the aging brain would have been necessary to confirm the study's findings and provided a higher understanding of the temporal relationships between 
αT stereoisomers.

ii) The study mainly focuses on the temporal cortex for αT stereoisomer analysis. While justified by similarities in stereoisomer distributions across regions in other studies also quoted by the authors, it would be beneficial to confirm these findings in multiple brain regions.

Please comment.

Overall, the manuscript provides a compelling and novel contribution to the field of the aging brain and vitamin E research, particularly through its focus on αT stereoisomers and their distinct roles in the aging brain.

Author Response

Please see attached Word document

Reviewer 2 Report

1. The observation that no correlation between the isomers and dementia (based on GDS) in the TC needs to be addressed in the Discussion since alphaT content is linked to better cognition. 

2. The role of BMI is not clear in the Discussion. Line 273, BMI is a crude indicator of overall health. Line 288, Overall while BMI is a significant factor in healthy ageing... The BMIs for the centenarians appear to be the normal range for this age-group. If BMI is removed from the analysis, the correlations are no longer significant. Is BMI relevant to this clinical scenario?

1. Lines 294-306. Largely speculative argument. How does this relate to extreme ageing?

2. Regarding study limitations on line 322. ".... profile determined from the TC to other brain regions is reasonable. Can we assume that young and old have similar Tc profiles?

Author Response

Please see attached Word document.

Round 2

Reviewer 2 Report

The authors have done a fine job addressing the comments. 

No comments here.